# Neural Speed Reading via Skim-RNN

**Minjoon Seo**[1,2*]     **Sewon Min**[3*]     **Ali Farhadi**[2,4,5]     **Hannaneh Hajishirzi**[2]
Clova AI Research, NAVER[1]     University of Washington[2]     Seoul National University[3]
Allen Institute for Artificial Intelligence[4]     XNOR.AI[5]
{minjoon, ali, hannaneh}@cs.washington.edu, shmsw25@snu.ac.kr

## Abstract

Inspired by the principles of speed reading, we introduce Skim-RNN, a recurrent neural network (RNN) that dynamically decides to update only a small fraction of the hidden state for relatively unimportant input tokens. Skim-RNN gives computational advantage over an RNN that always updates the entire hidden state. Skim-RNN uses the same input and output interfaces as a standard RNN and can be easily used instead of RNNs in existing models. In our experiments, we show that Skim-RNN can achieve significantly reduced computational cost without losing accuracy compared to standard RNNs across five different natural language tasks. In addition, we demonstrate that the trade-off between accuracy and speed of Skim-RNN can be dynamically controlled during inference time in a stable manner. Our analysis also shows that Skim-RNN running on a single CPU offers lower latency compared to standard RNNs on GPUs.

## 1 Introduction

Recurrent neural network (RNN) is a predominantly popular architecture for modeling natural language, where RNN sequentially 'reads' input tokens and outputs a distributed representation for each token. By recurrently updating the hidden state with an identical function, RNN inherently requires the same computational cost across time. While this requirement seems natural for some application domains, not all input token are equally important in many language processing tasks. For instance, in question answering, a rather efficient strategy would be to allocate less computation on irrelevant parts of the text (to the question) and only allow heavy computation on important parts.

Attention models (Bahdanau et al., 2014) compute the importance of the words relevant to the given task using an attention mechanism. They, however, do not focus on improving the efficiency of the inference. More recently, a variant of LSTMs (Yu et al., 2017) is introduced to improve inference efficiency through skipping multiple tokens at a given time step. In this paper, we introduce skim-RNN that takes advantage of 'skimming' rather than 'skipping' tokens. Skimming refers to the ability to decide to spend little time (rather than skipping) on parts of the text that does not affect the reader's main objective. Skimming typically gains trained human speed readers up to 4x speed up, occasionally with a bit of loss in the comprehension rates (Marcel Adam Just, 1987).

Inspired by the principles of human's speed reading, we introduce Skim-RNN (Figure 1), which makes a fast decision on the significance of each input (to the downstream task) and 'skims' through unimportant input tokens by using a smaller RNN to update only a fraction of the hidden state. When the decision is to 'fully read', Skim-RNN updates the entire hidden state with the default RNN cell. Since the hard decision function ('skim' or 'read') is non-differentiable, we use gumbel-softmax (Jang et al., 2017) to estimate the gradient of the function, instead of more traditional methods such as REINFORCE (policy gradient) (Williams, 1992). The switching mechanism between the two RNN cells enables Skim-RNN to reduce the total number of float operations (Flop reduction, or Flop-R) when the skimming rate is high, which often leads to faster inference on CPUs[1], a highly desirable goal for large-scale products and small devices.

---

[*]Equal contribution.

[1]Flop reduction does not necessarily mean equivalent speed gain. For instance, on GPUs, there will be no speed gain because of parallel computation. On CPUs, the gain will not be as high as the Flop-R due to overheads. See Section 4.3.

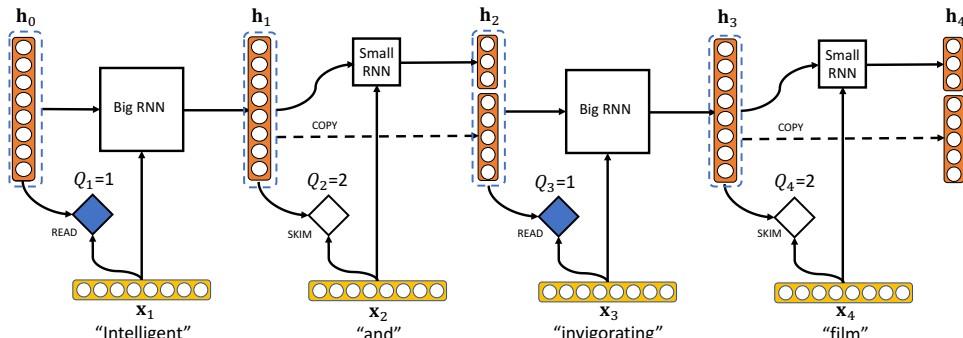

Figure 1: The schematic of Skim-RNN on a sample sentence from Stanford Sentiment Treebank: "intelligent and invigorating film". At time step 1, Skim-RNN makes the decision to *read* or *skim* $x_1$ by using Equation 1 on $h_0$ and $x_1$. Since 'intelligent' is an important word for sentiment, it decides to *read* (blue diamond) by obtaining a full-size hidden state with the big RNN and updating the entire previous hidden state. At time step 2, Skim-RNN decides to *skim* (empty diamond) the word 'and' by updating the first few dimensions of the hidden state using small RNN.

Skim-RNN has the same input and output interfaces as standard RNNs, so it can be conveniently used to speed up RNNs in existing models. This is in contrast to LSTM-Jump (Yu et al., 2017) that does not have outputs for the skipped time steps. Moreover, the speed of Skim-RNN can be dynamically controlled at inference time by adjusting the threshold for the 'skim' decision. Lastly, we show that skimming achieves higher accuracy compared to skipping the tokens, implying that paying some attention to unimportant tokens is better than completely ignoring (skipping) them.

Our experiments show that Skim-RNN attains computational advantage (float operation reduction, or Flop-R) over a standard RNN, with up to 3x reduction in computations while maintaining the same level of accuracy, on four text classification tasks and two question answering task. Moreover, for applications that are concerned with latency than throughput, Skim-RNN on a CPU can offer lower-latency inference time compared to to standard RNNs on GPUs (Section 4.3). Our experiments show that we achieve higher accuracy and/or computational efficiency compared to LSTM-jump and verify our intuition about the advantages of skimming compared to skipping.

## 2 RELATED WORK

**Fast neural networks.** As neural networks become widely integrated into real-world applications, making neural networks faster and lighter has drawn much attention in machine learning communities and industries recently. Mnih et al. (2014) perform hard attention instead of soft attention on image patches for caption generation, which reduces number of computations and memory usage. Han et al. (2016) compress a trained convolutional neural networks so that the model occupies less memory. Rastegari et al. (2016) approximate 32-bit float operations with single bit binary operations to substantially increase computational speed at the cost of little loss of precision. Odena et al. (2017) propose to change model behavior on per-input basis, which can decide to use less computation for simpler inputs.

More relevant work to ours are those that are specifically targeted for sequential data. LSTM-Jump (Yu et al., 2017) has the same goal as our model in that it aims to reduce the computational cost of recurrent neural networks. However, it is fundamentally different from skim-RNN in that it *skips* some input tokens while ours does not ignore any token and *skims* if the token is unimportant. Our experiments confirm the benefits of skimming compared to skipping in Figure 5. In addition, LSTM-Jump does not produce LSTM outputs for skipped tokens, which often means that it is nontrivial to replace a regular LSTM in existing models with LSTM-Jump, if the outputs of the LSTM (instead of just the last hidden state) is used. On the other hand, Skim-RNN emits a fixed-size output at every time step, so it is compatible with any RNN-based model. We also note the existence of Skip-LSTM (Campos et al., 2017), a recent, concurrent submission to ours that shares many characteristics with LSTM-Jump.

Variable Computation in RNN (VCRNN) (Jernite et al., 2017) is also concerned with dynamically controlling the computational cost of RNN. However, VCRNN only controls the number of units to update at each time step, while Skim-RNN contains multiple RNNs that "share" a common hidden state with different regions on which they operate (choosing which RNN to use at each time step). This has two important implications. First, the nested RNNs in Skim-RNN have their own weights and thus can be considered as independent agents that interact with each other through the shared state. That is, Skim-RNN updates the shared portion of the hidden state differently (by using different RNNs) depending on importance of the token, whereas the affected (first few) dimensions in VCRNN are identically updated regardless of the importance of the input. We argue that this capability of Skim-RNN could be a crucial advantage, as we demonstrate in Section 4. Second, at each time step, VCRNN needs to make a d-way decision (where $d$ is the hidden state size, usually hundreds), whereas Skim-RNN only requires binary decision. This means that computing exact gradient of VCRNN is even more intractable ($d^L$ vs $2^L$) than that of Skim-RNN, and subsequently the gradient estimation would be harder as well. We conjecture that this results in a higher variance in the performance of VCRNN per training, which we also discuss in Section 4.

Choi et al. (2017) use a CNN-based sentence classifier, which can be efficiently computed with GPUs, to select the most relevant sentence(s) to the question among hundreds of candidates, and uses an RNN-based question answering model, which is relatively costly on GPUs, to obtain the answer from the selected sentence. The two models are jointly trained with REINFORCE (Williams, 1992). Skim-RNN is inherently different from the model in that ours is generic (replaces RNN) and is not specifically for question answering, and Choi et al. (2017) the model focuses on reducing GPU-time (maximizing parallelization), while ours focuses on reducing CPU-time (minimizing Flop).

Johansen et al. (2017) have shown that, for sentiment analysis, it is possible to cheaply determine if entire sentence can be correctly classified with a cheap bag-of-word model or needs a more expensive LSTM classifier. Again, Skim-RNN is intrinsically different from their approach in that it makes a single, static decision on which model to use on the entire example.

**Attention.** Modeling human's attention while reading has been studied in the field of cognitive psychology (Reichle et al., 2003). Neural attention mechanism has been also widely employed and proved to be essential for many language tasks (Bahdanau et al., 2014), allowing the model to focus on specific parts of of the text. Nevertheless, it is important to note the distinction from Skim-RNN that the neural attention mechanism is soft (differentiable) and is not intended for faster inference. More recently, Hahn & Keller (2016) have modeled the human reading pattern with neural attention in an unsupervised learning approach, leading to conclusion that there exists trade-off between a system's performance in a given reading-based task and the speed of reading.

**RNNs with hard decisions.** Our model is relevant to several recent works that incorporate hard decisions within recurrent neural networks (Kong et al., 2016). Dyer et al. (2016) uses RNN for transition-based dependency parsing. At each time step, the RNN unit decides between three possible choices. The architecture does not suffer from the intractability of computing the gradients, because the decision is supervised at every time step. Chung et al. (2017) dynamically construct multiscale RNN by making a hard binary decision on whether to update hidden state of each layer at each time step. In order to handle the intractability of computing the gradient, they use straight-through estimation (Bengio et al., 2013) with slope annealing, which can be considered as an alternative method to Gumbel-softmax reparameterization.

## 3 MODEL

Skim-RNN unit consists of two RNN cells, default (big) RNN cell of hidden state size $d$ and small RNN cell of hidden state size $d'$, where $d$ and $d'$ are hyperparameters defined by the user and $d' \ll d$. Each RNN cell has its own weight and bias, and it can be any variant of RNN, such as GRU and LSTM. The core idea of the model is that the Skim-RNN dynamically makes the decision at each time step whether to use the big RNN (if the current token is important), or to *skim* by using the small RNN (if the current token is unimportant). Skipping a token can be implemented by setting $d'$, the size of the small RNN, equal to zero. Since small RNN requires less number of float operations than big RNN, the model is faster than big RNN alone while obtaining similar or better results than the big RNN alone. Later in Section 4, we will measure the speed effectiveness of Skim-RNN via three

criteria: skim rate (how many words are skimmed), number of float operations, and benchmarked speed on several platforms. Figure 1 depicts the schematic of Skim-RNN on a short word sequence.

We first describe the desired inference model of Skim-RNN to be learned in Section 3.1. The input to and the output of Skim-RNN are equivalent to that of a regular RNN: a varying-length sequence of vectors go in, and an equal-length sequence of output vectors come out. We model the hard decision of *skimming* at each time step with a stochastic multinomial variable. Note that obtaining the exact gradient is intractable as the sequence becomes longer, and the loss is not differentiable due to hard argmax; hence, in Section 3.2, we reparameterize the stochastic distribution with Gumbel-softmax (Jang et al., 2017) to approximate the inference model with a fully-differentiable function, which can be efficiently trained with stochastic gradient descent.

## 3.1 INFERENCE

At each time step $t$, Skim-RNN unit takes the input $\mathbf{x}_t \in \mathbb{R}^d$ and the previous hidden state $\mathbf{h}_{t-1} \in \mathbb{R}^d$ as its arguments, and outputs the new state $\mathbf{h}_t$.[2] Let $k$ represent the number of choices for the hard decision at each time step. In Skim-RNNs, $k = 2$ since it either fully reads or *skims*. In general, although not explored in this paper, one can have $k > 2$ for multiple degrees of skimming.

We model the decision making process with a multinomial random variable $Q_t$ over the probability distribution of choices $\mathbf{p}_t$. We model $\mathbf{p}_t$ with

$$\mathbf{p}_t = \mathrm{softmax}(\alpha(\mathbf{x}_t, \mathbf{h}_{t-1})) = \mathrm{softmax}(\mathbf{W}[\mathbf{x}_t; \mathbf{h}_{t-1}] + \mathbf{b}) \in \mathbb{R}^k, \tag{1}$$

where $\mathbf{W} \in \mathbb{R}^{k \times 2d}$ and $b \in \mathbb{R}^k$ are weights to be learned, and $[;]$ indicates row concatenation. Note that one can define $\alpha$ in a different way (e.g., the dot product between $\mathbf{x}_t$ and $\mathbf{h}_{t-1}$), as long as its time complexity is strictly less than $O(d^2)$ to gain computational advantage. For the ease of explanation, let the first element of the vector, $\mathbf{p}_t^1$, indicate the probability for fully reading, and the second element, $\mathbf{p}_t^2$, indicate the probability for skimming. Now we define the random variable $Q_t$ to make the decision to skim ($Q_t = 2$) or not ($Q_t = 1$), by sampling $Q_t$ from the probability distribution $\mathbf{p}_t$.

$$Q_t \sim \mathrm{Multinomial}(\mathbf{p}_t), \tag{2}$$

which means $Q_t = 1$ and $Q_t = 2$ will be sampled with the probability of $\mathbf{p}_t^1$ and $\mathbf{p}_t^2$, respectively. If $Q_t = 1$, then the unit applies a standard, full RNN on the input and the previous hidden state to obtain the new hidden state. If $Q_t = 2$, then the unit applies a smaller RNN to obtain a small hidden state, which replaces only a portion of the previous hidden state. More formally,

$$\mathbf{h}_t = \begin{cases} f(\mathbf{x}_t, \mathbf{h}_{t-1}), & \text{if } Q_t = 1, \\ [f'(\mathbf{x}_t, \mathbf{h}_{t-1}); \mathbf{h}_{t-1}(d'+1:d)], & \text{if } Q_t = 2, \end{cases} \tag{3}$$

where $f$ is a full RNN with $d$-dimensional output, while $f'$ is a smaller RNN with $d'$-dimensional output, where $d' \ll d$, and $(:)$ is vector slicing. Note that $f$ and $f'$ can be any variant of RNN such as GRU and LSTM[3]. The main computational advantage of the model is that, if $d' \ll d$, then whenever the model decides to skim, it requires $O(d'd)$ computations, which is substantially less than $O(d^2)$. Also, as a side effect, the last $d - d'$ dimensions of the hidden state are less frequently updated, which we hypothesize to be a nontrivial factor for improved accuracy in some datasets (Section 4).

## 3.2 TRAINING

Since the loss is a random variable that depends on the random variables $Q_t$, we minimize the expected loss with respect to the distribution of the variables.[4] Suppose that we define the loss function to be minimized conditioned on a particular sequence of decisions, $L(\theta; Q)$ where $Q = Q_1 \dots Q_T$ is a sequence of decisions with length $T$. Then the expectation of the loss function over the distribution of the sequence of the decisions is

$$\mathbb{E}_{Q_t \sim \mathrm{Multinomial}(\mathbf{p}_t)}[L(\theta)] = \sum_Q L(\theta; Q) P(Q) = \sum_Q L(\theta; Q) \prod_j \mathbf{p}_j^{Q_j}. \tag{4}$$

---

[2]We assume both input and hidden state are $d$-dimensional for brevity, but our arguments are valid for different sizes as well.

[3]Since LSTM cell has two outputs, hidden state ($\mathbf{h}_t$) and memory ($\mathbf{c}_t$), the slicing and concatenation in Equation 3 is applied for each output.

[4]An alternative view is that, if we let $Q_t = \arg\max(\mathbf{p}_t)$ instead of sampling, which we do during inference for deterministic outcome, then the loss is non-differentiable due to the argmax operation (hard decision).

In order to exactly compute $\nabla \mathbb{E}_{Q_t}[L(\theta)]$, one needs to enumerate all possible $Q$, which is intractable (exponentially increases with the sequence length). It is possible to approximate the gradients with REINFORCE (Williams, 1992), which is an unbiased estimator, but it is known to have a high variance. We instead use gumbel-softmax distribution (Jang et al., 2017) to approximate Equation 2, $\mathbf{r}_t \in \mathbb{R}^k$ (same size as $\mathbf{p}_t^i$), which is fully differentiable. Hence the back-propagation can now efficiently flow to $\mathbf{p}_t$ without being blocked by the stochastic variable $Q_t$, and the approximation can arbitrarily approach to $Q_t$ by controlling hyperparameters. The reparameterized distribution is obtained by

$$\mathbf{r}_t^i = \frac{\exp((\log(\mathbf{p}_t^i) + g_t^i)/\tau)}{\sum_j \exp((\log(\mathbf{p}_t^j) + g_t^j)/\tau)} \tag{5}$$

where $g_t^i$ is an independent sample from $\mathrm{Gumbel}(0, 1) = -\log(-\log(\mathrm{Uniform}(0, 1)))$ and $\tau$ is the *temperature* (hyperparameter). We relax the conditional statement of Equation 3 by rewriting $\mathbf{h}_t$

$$\mathbf{h}_t = \sum_i \mathbf{r}_t^i \tilde{\mathbf{h}}_t^i \tag{6}$$

where $\tilde{\mathbf{h}}_t^i$ is the candidate hidden state if $Q_t = i$. That is,

$$\begin{aligned}
\tilde{\mathbf{h}}_t^1 &= f(\mathbf{x}_t, \mathbf{h}_{t-1}) \\
\tilde{\mathbf{h}}_t^2 &= [f'(\mathbf{x}_t, \mathbf{h}_{t-1}); \mathbf{h}_{t-1}(d'+1:d)]
\end{aligned} \tag{7}$$

as shown in Equation 3. Note that Equation 6 approaches Equation 3 as $\mathbf{r}_t^i$ approaches to be a one-hot vector. Jang et al. (2017) have shown that $r_t$ becomes more discrete and approaches the distribution of $Q_t$ as $\tau \to 0$. Hence we start from a high temperature (smoother $\mathbf{r}_t$) value and slowly decreases it.

Lastly, in order to encourage the model to *skim* when possible, in addition to minimizing the main loss function ($L(\theta)$), which is application-dependent, we also jointly minimize the arithmetic mean of the negative log probability of *skimming*, $\frac{1}{T}\sum \log(\mathbf{p}_t^2)$, where $T$ is the sequence length. We define the final loss function by

$$L'(\theta) = L(\theta) + \gamma \frac{1}{T} \sum_t -\log(\mathbf{p}_t^2), \tag{8}$$

where $\gamma$ is a hyperparameter to control the ratio between the two terms.

## 4 EXPERIMENTS

| Dataset | task type | answer type | Number of examples | Avg. Len | vocab size |
|---|---|---|---|---|---|
| SST | Sentiment Analysis | Pos/Neg | 6,920 / 872 / 1,821 | 19 | 13,750 |
| Rotten Tomatoes | Sentiment Analysis | Pos/Neg | 8,530 / 1,066 / 1,066 | 21 | 16,259 |
| IMDb | Sentiment Analysis | Pos/Neg | 21,143 / 3,857 / 25,000 | 282 | 61,046 |
| AGNews | News classification | 4 categories | 101,851 / 18,149 / 7,600 | 43 | 60,088 |
| CBT-NE | Question Answering | 10 candidates | 108,719 / 2,000 / 2,500 | 461 | 53,063 |
| CBT-CN | Question Answering | 10 candidates | 120,769 / 2,000 / 2,500 | 500 | 53,185 |
| SQuAD | Question Answering | span from context | 87,599 / 10,570 / - | 141 | 69,184 |

Table 1: Statistics and the examples of the datasets that Skim-RNN is evaluated on. SST refers to Stanford Sentiment Treebank, SQuAD refers to Stanford Question Answering Dataset, CBT-NE refers to Named Entity dataset of Children Book Test, and CBT-CN refers to Common Noun of CBT.

We evaluate the effectiveness of Skim-RNN in terms of accuracy and float operation reduction (Flop-R) on four classification tasks and a question answering task. These language tasks have been chosen because they do not require one's full attention to every detail of the text, but rather ask for capturing the high-level information (classification) or focusing on specific portion (QA) of the text, which is more appropriate for the principle of speed reading[5].

We start with classification tasks (Section 4.1) and compare Skim-RNN against standard RNN, LSTM-Jump (Yu et al., 2017), and VCRNN (Jernite et al., 2017), which have a similar goal to ours. Then we evaluate and analyze our system in a well-studied question answering dataset, Stanford

---

[5] 'Speed reading' would not be appropriate for many language tasks. For instance, in translation task, one would not skim through the text because most input tokens are crucial for the task.

| LSTM Model | $d'/\gamma$ | SST | | | | Rotten Tomatoes | | | | IMDb | | | | AGNews | | | |
|---|---|---|---|---|---|---|---|---|---|---|---|---|---|---|---|---|---|
| | | Acc | Sk | Flop-r | Sp | Acc | Sk | Flop-r | Sp | Acc | Sk | Flop-r | Sp | Acc | Sk | Flop-r | Sp |
| Standard | | 86.4 | - | 1.0x | 1.0x | 82.5 | - | 1.0x | 1.0x | 91.1 | - | 1.0x | 1.0x | 93.5 | - | 1.0x | 1.0x |
| Skim | 5/0.01 | **86.4** | 58.2 | 2.4x | 1.4x | **84.2** | 52.0 | 2.1x | 1.3x | 89.3 | 79.2 | 4.7x | 2.1x | **93.6** | 30.3 | 1.4x | 1.0x |
| Skim | 10/0.01 | 85.8 | 61.1 | 2.5x | 1.5x | 82.5 | 58.5 | 2.4x | 1.4x | **91.2** | 83.9 | 5.8x | 2.3x | 93.5 | 33.7 | 1.5x | 1.0x |
| Skim | 5/0.02 | 85.6 | 62.3 | 2.6x | 1.5x | 81.8 | 63.7 | 2.7x | 1.5x | 88.7 | 63.2 | 2.7x | 1.5x | 93.3 | 36.4 | 1.6x | 1.0x |
| Skim | 10/0.02 | **86.4** | 68.0 | 3.0x | 1.7x | 82.5 | 63.0 | 2.6x | 1.5x | 90.9 | 90.7 | 9.5x | 2.7x | 92.5 | 10.6 | 1.1x | 0.8x |
| LSTM-Jump | - | - | - | - | - | 79.3 | - | - | 1.6x | 89.4 | - | - | 1.6x | 89.3 | - | - | 1.1x |
| VCRNN | | 81.9 | - | 2.6x | - | - | - | - | - | - | - | - | - | - | - | - | - |
| SOTA | | 89.5 | - | - | - | 83.4 | - | - | - | 94.1 | - | - | - | 93.4 | - | - | - |

Table 2: Text classification results on SST, Rotten Tomatoes, IMDb and AGNews. Results by standard LSTM, Skim-LSTM, LSTM-Jump (Yu et al., 2017), VCRNN (Jernite et al., 2017) and state of the art (SOTA). Evaluation metrics are accuracy (Acc), skimming rate in % (Sk), reduction rate in the number of floating point operations (Flop-r) compared to standard LSTM, and benchmarked speed up rate (Sp) compared to standard LSTM. We use the hidden size of 100 by default. SOTAs are from Kokkinos & Potamianos (2017), Miyato et al. (2017), Miyato et al. (2017) and Zhang et al. (2015), respectively.

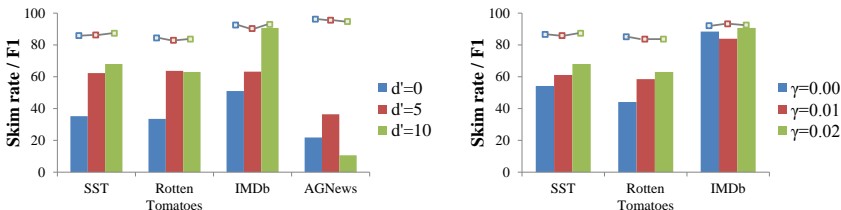

Figure 2: Analyzing the effect of small hidden state size, d' (left) and $\gamma$ (right) on skim rate; ($d = 100$, $d' = 10$, and $\gamma = 0.02$ are default values).

Question Answering Dataset (SQuAD) (Section 4.2). Since LSTM-Jump does not report on this dataset, we simulate 'skipping' by not updating the hidden state when the decision is to 'skim', and show that skimming yields better accuracy-speed trade-off than skipping. We defer the results of Skim-RNN on Children Book Test to Appendix B.

**Evaluation Metrics.** We measure the accuracy for the the classification task (Acc) and the F1 and exact match (EM) scores of the correct span for the question answering task. We evaluate the computational efficiency with skimming rate (Sk) i.e., how frequently words are skimmed, and reduction in float operations (Flop-R). We also report benchmarked speed gain rate (compared to standard LSTM) of classification tasks and CBT since LSTM-Jump does not report Flop reduction rate (See Section 4.3 for how the benchmark is performed). Note that LSTM-Jump measures speed gain based on GPU while ours is measured based on CPU.

## 4.1 TEXT CLASSIFICATION

In a language classification task, the input is a sequence of words and the output is the vector of categorical probabilities. Each word is embedded into a $d$-dimensional vector. We initialize the vector with GloVe (Pennington et al., 2014) and use those as the inputs for LSTM (or Skim-LSTM). We make a linear transformation on the last hidden state of the LSTM and then apply softmax function to obtain the classification probabilities. We use Adam (Kingma & Ba, 2015) for optimization, with initial learning rate of 0.0001. For Skim-LSTM, $\tau = \max(0.5, \exp(-rn))$ where $r = 1e - 4$ and $n$ is the global training step, following Jang et al. (2017). We experiment on different sizes of big LSTM ($d \in \{100, 200\}$) and small LSTM ($d' \in \{5, 10, 20\}$) and the ratio between the model loss and the skim loss ($\gamma \in \{0.01, 0.02\}$) for Skim-LSTM. We use batch size of 32 for SST and Rotten Tomatoes, and 128 for others. For all models, we stop early when the validation accuracy does not increase for 3000 global steps.

**Results.** Table 2 shows the accuracy and the computational cost of our model compared with standard LSTM, LSTM-Jump (Yu et al., 2017), and VCRNN (Jernite et al., 2017). First, Skim-LSTM has a significant reduction in number of float operations compared to LSTM, as indicated by 'Flop-R'. When benchmarked on Python ('Sp' column), we observe a nontrivial speed up. We expect that the gain can be further maximized when implemented with lower level language that has smaller overhead. Second, our model outperforms standard LSTM and LSTM-Jump across all tasks, and its accuracy is better than or close to that of RNN-based state of the art models, which are often

| Positive | I **liked** this movie, not because Tom Selleck was in it, but **because** it was a **good** story about baseball and it also had a semi-over **dramatized** view of some of the issues that a BASEBALL player coming to the end of their time in Major League sports must face. I also **greatly enjoyed** the cultural differences in American and Japanese baseball and the small facts on how the games are played differently. **Overall**, it is a **good movie** to watch on Cable TV or rent on a cold winter's night and watch about the "Dog Day's" of summer and know that spring training is only a few months away. A **good** movie for a baseball fan as well as a good "DATE" movie. Trust me on that one! *Wink* |
|----------|---|
| Negative | **No**! **no** - **No** - **NO**! My **entire** being is **revolting** against this **dreadful** remake of a classic movie. I **knew** we were heading for trouble from the moment Meg Ryan appeared on screen with her **ridiculous** hair and clothing - literally looking like a scarecrow in that garden she was digging. Meg Ryan playing Meg Ryan - how **tiresome** is that?! And it **got worse** ... so much **worse**. The **horribly cliché** lines, the stock characters, the increasing sense I was **watching** a spin-off of "The First Wives Club" and the ultimate **hackneyed** schtick in the delivery room. How many times have I seen this movie? Only once, but it **feel** like a dozen times - **nothing** original or fresh about it. For shame! |

Table 3: A positive and a negative review from IMDb dataset. Black-colored words are skimmed (used smaller LSTM, $d' = 10$), while blue-colored words are fully read (used bigger LSTM, $d = 200$).

specifically designed for these tasks. We hypothesize the accuracy improvement over LSTM could be due to the increased stability of the hidden state, as the majority of the hidden state is not updated when skimming. Figure 2 shows the effect of varying the size of the small hidden state as well as the parameter $\gamma$ on the accuracy and computational cost.

Table 3 shows an example from IMDb dataset, where Skim-RNN with $d = 200$, $d' = 10$, and $\gamma = 0.01$ correctly classifies it with high skimming rate (92%). The black words are skimmed, and blue words are fully read. As expected, the model skims through unimportant words, including prepositions, and latently learns to only carefully read the important words, such as 'liked', 'dreadful', and 'tiresome'.

## 4.2 QUESTION ANSWERING

In Stanford Question Answering Dataset, the task is to locate the answer span for a given question in a context paragraph. We evaluate the effectiveness of Skim-RNN for SQuAD with two different models: LSTM+Attention and BiDAF (Seo et al., 2017). The first model is inspired by most current QA systems consisting of multiple LSTM layers and an attention mechanism. The model is complex enough to reach reasonable accuracy on the dataset, and simple enough to run well-controlled analyses for the Skim-RNN. The details of the model are described in Appendix A.1. The second model is an open-source model designed for SQuAD, which is studied to mainly show that Skim-RNN could replace RNN in existing complex systems.

**Training.** We use Adam and initial learning rate of 0.0005. For stable training, we pretrain with standard LSTM for the first 5k steps , and then finetune with Skim-LSTM (Section A.2 shows different pretraining schemas). Other hyperparameter setup follows that of classification in Section 4.1.

**Results.** Table 4 (above double line) shows the accuracy (F1 and EM) of LSTM+Attention and Skim-LSTM+Attention models as well as VCRNN (Jernite et al., 2017). We observe that the skimming models achieve higher or similar F1 score to those of the default non-skimming models (LSTM+Att) while attaining the reduction in computational cost (Flop-R) by more than 1.4 times. Moreover, decreasing layers (1 layer) or hidden size (d=5) improves Flop-R, but significantly decreases the accuracy (compared to skimming). Table 4 (below double line) demonstrates that replacing LSTM with Skim-LSTM in an existing complex model (BiDAF) stably gives reduced computational cost without losing much accuracy (only 0.2% drop from 77.3% of BiDAF to 77.1% of Sk-BiDAF with $\gamma = 0.001$).

Figure 3 shows the skimming rate of different layers of LSTM with varying values of $\gamma$ in LSTM+Att model. There are four points on the axis of the figures associated with two forward and two backward layers of the model. We see two interesting trends here. First, the skimming rate of the second layers (forward and backward) are higher than that of the first layer across different gamma values. A possible explanation for this trend is that the model is more confident about which tokens are important at the second layer. Second, higher $\gamma$ value leads to higher skimming rate, which agrees with its intended functionality.

Figure 4 shows F1 score of LSTM+Attention model using standard LSTM and Skim LSTM, sorted in ascending order by Flop-R. While models tend to perform better with larger computational cost, Skim LSTM (Red) outperforms standard LSTM (Blue) with comparable computational cost. We

| Model | $\gamma$ | F1 | EM | Sk | Flop-r |
|---|---|---|---|---|---|
| LSTM+Att (1 layer) | - | 73.3 | 63.9 | - | 1.3x |
| LSTM+Att ($d = 50$) | - | 74.0 | 64.4 | - | 3.6x |
| LSTM+Att | - | 75.5 | **67.0** | - | 1.0x |
| Sk-LSTM+Att ($d' = 0$) | 0.1 | **75.7** | 66.7 | 37.7 | 1.4x |
| Sk-LSTM+Att ($d' = 0$) | 0.2 | 75.6 | 66.4 | 49.7 | 1.6x |
| Sk-LSTM+Att | 0.05 | 75.5 | 66.0 | 39.7 | 1.4x |
| Sk-LSTM+Att | 0.1 | 75.3 | 66.0 | 56.2 | 1.7x |
| Sk-LSTM+Att | 0.2 | 75.0 | 66.0 | 76.4 | 2.3x |
| VCRNN | - | 74.9 | 65.4 | - | 1.0x |
| BiDAF ($d = 30$) | - | 74.6 | 64.0 | - | 9.1x |
| BiDAF ($d = 50$) | - | 75.7 | 65.5 | - | 3.7x |
| BiDAF | - | **77.3** | **67.7** | - | 1.0x |
| Sk-BiDAF | 0.01 | 76.9 | 67.0 | 74.5 | 2.8x |
| Sk-BiDAF | 0.001 | 77.1 | 67.4 | 47.1 | 1.7x |
| SOTA (Wang et al., 2017) | | 79.5 | 71.1 | - | - |

Table 4: Results on Stanford Question Answering Dataset (SQuAD), using LSTM+Attention (2 layers of LSTM, $d = 100$, $d' = 20$ by default) and BiDAF ($d = 100$, $d' = 50$ by default).

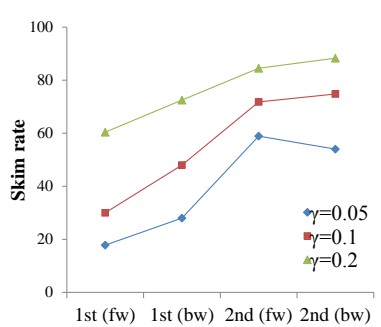

Figure 3: Skim rate of LSTMs in LSTM+Att model. Two layers of forward and backward LSTMs are shown (total count of 4), with $d = 100$, $d' = 20$.

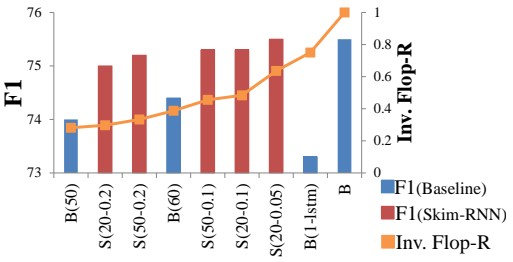

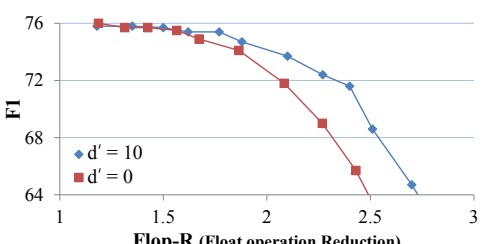

Figure 4: F1 score of standard LSTM with varying configurations (Blue) and Skim LSTM with varying configurations (Red), both sorted together in ascending order by the inverse of Flop-R (Orange). $d = 100$ by default. Numbers inside B refer to $d$, and numbers inside S refer to $d', \gamma$.

Figure 5: Trade-off between F1 score and Flop-R obtained by adjusting the threshold for the skim (or skip) decision. Blue line is a skimming model with $d' = 10$, and red line is a skipping model ($d' = 0$). The gap between the lines shows the advantage of skimming over skipping.

also observe that the F1 score of Skim-LSTM is more stable across different configurations and computational cost. Moreover, increasing the value of $\gamma$ for Skim-LSTM gradually increases skipping rate and Flop-R, while it also leads to reduced accuracy.

**Controlling skim rate.** An important advantage of Skim-RNN is that the skim rate (and thus computational cost) can be dynamically controlled at inference time by adjusting the threshold for 'skim' decision probability $\mathbf{p}_t^1$ (Equation 1). Figure 5 shows the trade-off between the accuracy and computational cost for two settings, confirming the importance of skimming ($d' > 0$) compared to skipping ($d' = 0$).

**Visualization.** Figure 6 shows an example from SQuAD and visualizes which words Skim-LSTM ($d = 100, d' = 20$) reads (red) and skims (white). As expected, the model does not skim when the input seems to be relevant to answering the question. In addition, LSTM in second layer skims more than that in the first layer mainly because the second layer is more confident about the importance of each token, as shown in Figure 6. More visualizations are shown in in Appendix C.

## 4.3 RUNTIME BENCHMARKS

Here we briefly discuss the details of the runtime benchmarks for LSTM and Skim-LSTM, which allow us to estimate the speed up of Skim-LSTM-based models in our experiments (corresponding to 'Sp' in Table 2). We assume CPU-based benchmark by default, which has direct correlation with the number of float operations (Flop)[6]. As mentioned previously, the speed-up results in Table 2 (as well as Figure 7 below) are benchmarked using Python (NumPy), instead of popular frameworks such as TensorFlow or PyTorch. In fact, we have benchmarked the speed of Length-100 LSTM with $d = 100$

---

[6]Speed up on GPUs hugely depends on parallelization, which is not relevant to our contribution.

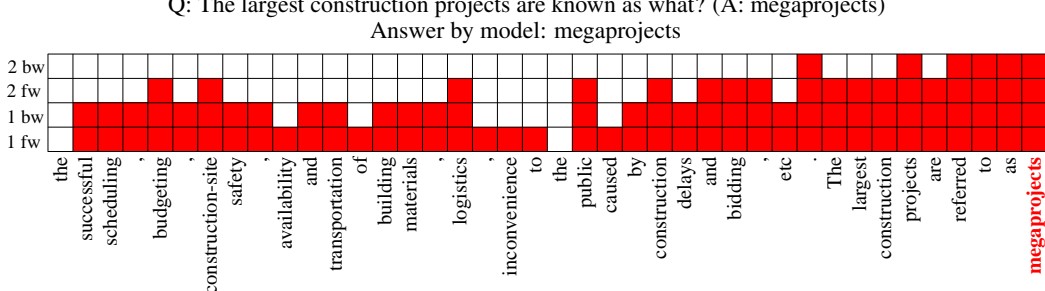

Figure 6: Reading (red) and skimming (white) decisions in four LSTM layers (two for forward and two for backward) of Skim-LSTM+Attention model. We see that the second layer skims more, implying that the second layer is more confident about which tokens are important.

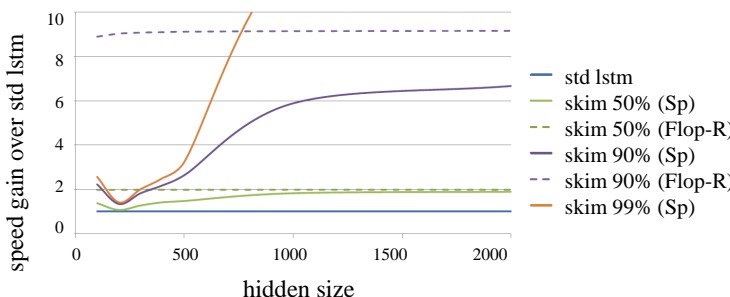

Figure 7: Speed up rate of Skim-LSTM (vs LSTM) with varying skimming rates and hidden state sizes.

(batch size = 1) in all three frameworks on a single thread of CPU (averaged over 100 trials), and have observed that NumPy is 1.5 and 2.8 times faster than TensorFlow and PyTorch.[7] This seems to be mostly due to the fact that the frameworks are primarily (optimized) for GPUs and they have larger overhead than NumPy that they cannot take much advantage of reducing the size of the hidden state of the LSTM below 100.

Figure 7 shows the relative speed gain of Skim-LSTM compared to standard LSTM with varying hidden state size and skim rate. We use NumPy, and the inferences are run on a single thread of CPU. We also plot the ratio between the reduction of the number of float operations (Flop-R) of LSTM and Skim-LSTM. This can be considered as a theoretical upper bound of the speed gain on CPUs. We note two important observations. First, there is an inevitable gap between the actual gain (solid line) and the theoretical gain (dotted line). This gap will be larger with more overhead of the framework, or more parallelization (e.g. multithreading). Second, the gap decreases as the hidden state size increases because the the overhead becomes negligible with very large matrix operations. Hence, the benefit of Skim-RNN will be greater for larger hidden state size.

**Latency.** A modern GPU has much higher throughput than a CPU with parallel processing. However, for small networks, the CPU often has lower latency than the GPU. Comparing between NumPy with CPU and TensorFlow with GPU (Titan X), we observe that the former has 1.5 times lower latency (75 μs vs 110 μs per token) for LSTM of $d = 100$. This means that combining Skim-RNN with CPU-based framework can lead to substantially lower latency than GPUs. For instance, Skim-RNN with CPU on IMDb has 4.5x lower latency than a GPU, requiring only 29 μs per token on average.

## 5 CONCLUSION

We present Skim-RNN, a recurrent neural network that can dynamically decide to use the big RNN (read) or the small RNN (skim) at each time step, depending on the importance of the input. While

---

[7]NumPy's speed becomes similar to that of TensorFlow and PyTorch at $d = 220$ and $d = 700$, respectively. At larger hidden size, NumPy becomes slower.

Skim-RNN has significantly lower computational cost than its RNN counterpart, the accuracy of Skim-RNN is still on par with or better than standard RNNs, LSTM-Jump, and VCRNN. Since Skim-RNN has the same input and output interface as an RNN, it can easily replace RNNs in existing applications. We also show that a Skim-RNN can offer better latency results on a CPU compared to a standard RNN on a GPU. Future work involves using Skim-RNN for applications that require much higher hidden state size, such as video understanding, and using multiple small RNN cells for varying degrees of skimming.

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

## A  MODELS AND TRAINING DETAILS ON SQuAD

### A.1  LSTM+ATTENTION DETAILS

Let $\mathbf{x}_t$ and $\mathbf{q}_i$ be the embeddings of $t$-th context word and $i$-th question word, respectively. We first obtain the $d$-dimensional representation of the entire question by computing the weighted average of the question word vectors. We obtain $\mathbf{a}_t = \mathrm{softmax}_i(\mathbf{w}^\top[\mathbf{x}_t; \mathbf{q}_i; \mathbf{x}_t \circ \mathbf{q}_t])$ and $\mathbf{u}_t = \sum_i \mathbf{a}_t\mathbf{q}_i$, where $\mathbf{w} \in \mathbb{R}^{3d}$ is a trainable weight vector and $\circ$ is element-wise multiplication. Then the input to the (two layer) Bidirectional LSTMs will be $[\mathbf{x}_t; \mathbf{u}_t; \mathbf{x}_t \circ \mathbf{u}_t] \in \mathbb{R}^{3d}$. We use the outputs of the second layer LSTM to independently predict the start index and the end index of the answer. We obtain the logits (to be softmaxed) of the start and the end index distributions from the weighted average of the outputs (the weights are learned and different for the start and the end). We minimize the sum of the negative log probabilities of the correct start/end indices.

### A.2  USING PRE-TRAINED MODEL

**(a) No pretrain**   We train Skim LSTM from scratch. It has unstable skim rates, which are often too high or too low, and have very different skim rate in forward and backward direction of LSTM, with a significant loss in performance.

**(b) Full pretrain**   We finetune Skim LSTM from fully pretrained standard LSTM (F1 75.5, global step 18k). As we finetune the model, performance decreases and skim rate increases.

**(c) Half pretrain**   We finetune Skim LSTM from partially-pretrained standard LSTM (F1 70.7, pretraining stopping at 5k steps). Performance and skim rate increase together during training.

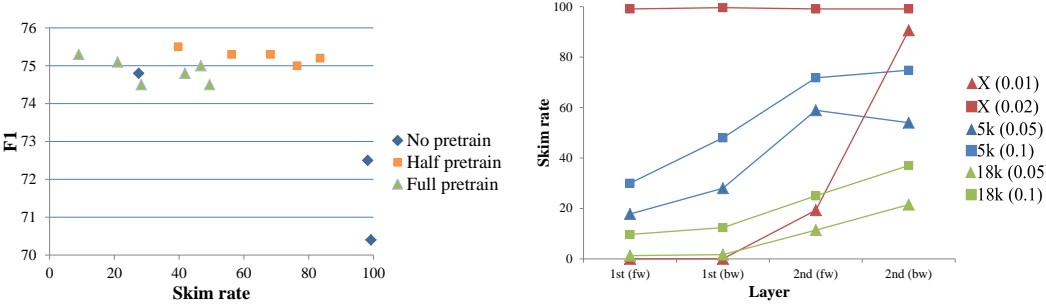

Figure 8: F1 score and skim rate when using different pretraining schemas. Models with half-pretrained model (Yellow) outperforms models with no pretrained model (Blue) or fully pretrained model (Green), both in F1 score and skim rate.

Figure 9: Skim rate in different layers of LSTM, when using different pretrained models. All models have hidden size of 100 and small hidden size of 20. Number inside parenthesis indicates $\gamma$. Models with no pretrained model (Red) have unstable skim rates.

## B  EXPERIMENTS ON CHILDREN BOOK TEST

In Children Book Test, the input is a sequence of 21 sentences, where the last sentence has one missing word (i.e. cloze test). The system needs to predict the missing word, which is one of ten provided candidates. Following LSTM-Jump (Yu et al., 2017), we use a simple LSTM-based QA model that help us to compare against LSTM and LSTM-Jump. We use single-layer LSTM on the embeddings of the inputs and use the last hidden state of the LSTM for the classification, where the output distribution is obtained by performing softmax on the dot product between the embedding of each answer candidate and the hidden state. We minimize the negative log probability of the correct answers. We follow the same hyperparameter setup and evaluation metrics from that of Section 4.1.

**Results.**   In Table 5, we first note that Skim-LSTM obtain better results than standard LSTM and LSTM-Jump. As discussed in Section 4.1, we hypothesize that the increase in accuracy could be due to the stabilization of the recurrent hidden state over a long distance. Second, using Skim-LSTM, we see up to 72.2% skimming rate, 3.6x reduction in the number of floating point operations and 2.2x

reduction in actual benchmarked time on NumPy with reasonable accuracies. Lastly, we note that LSTM, LSTM-Jump, and Skim-LSTM are all significantly lower than the state of the art models, which consist of a wide variety of components such as attention mechanism which are often crucial for question answering models.

| LSTM | config | CBT-NE | | | | CBT-CN | | | |
|---|---|---|---|---|---|---|---|---|---|
| Model | $d'/\gamma$ | Acc | Sk | Flop-r | Sp | Acc | Sk | Flop-r | Sp |
| Std | -/- | 49.0 | - | 1.0x | 1.0x | 54.9 | - | 1.0x | 1.0x |
| Skim | 10/0.01 | 50.9 | 35.8 | 1.6x | 1.3x | **56.3** | 43.7 | 1.8x | 1.5x |
| Skim | 10/0.02 | **51.4** | 72.2 | 3.6x | 2.3x | 51.4 | 86.5 | 7.1x | 3.3x |
| Skim | 20/0.01 | 36.4 | 98.8 | 50.0x | 1.3x | 38.0 | 99.1 | 50.0x | 1.3x |
| Skim | 20/0.02 | 50.0 | 70.5 | 3.3x | 1.2x | 54.5 | 54.1 | 2.1x | 1.1x |
| LSTM-Jump (Yu et al., 2017) | | 46.8 | - | - | 3.0x | 49.7 | - | - | 6.1x |
| SOTA (Yang et al., 2017) | | 75.0 | - | - | - | 72.0 | - | - | - |

Table 5: Question answering experiments with standard LSTM, Skim-LSTM, LSTM-Jump (Yu et al., 2017) and state of the art (SOTA) on NE and CN parts of Children Book Test (CBT). Hidden state sizes of all models are 200, except for LSTM-Jump, which used hidden state size of 512.

## C VISUALIZATION

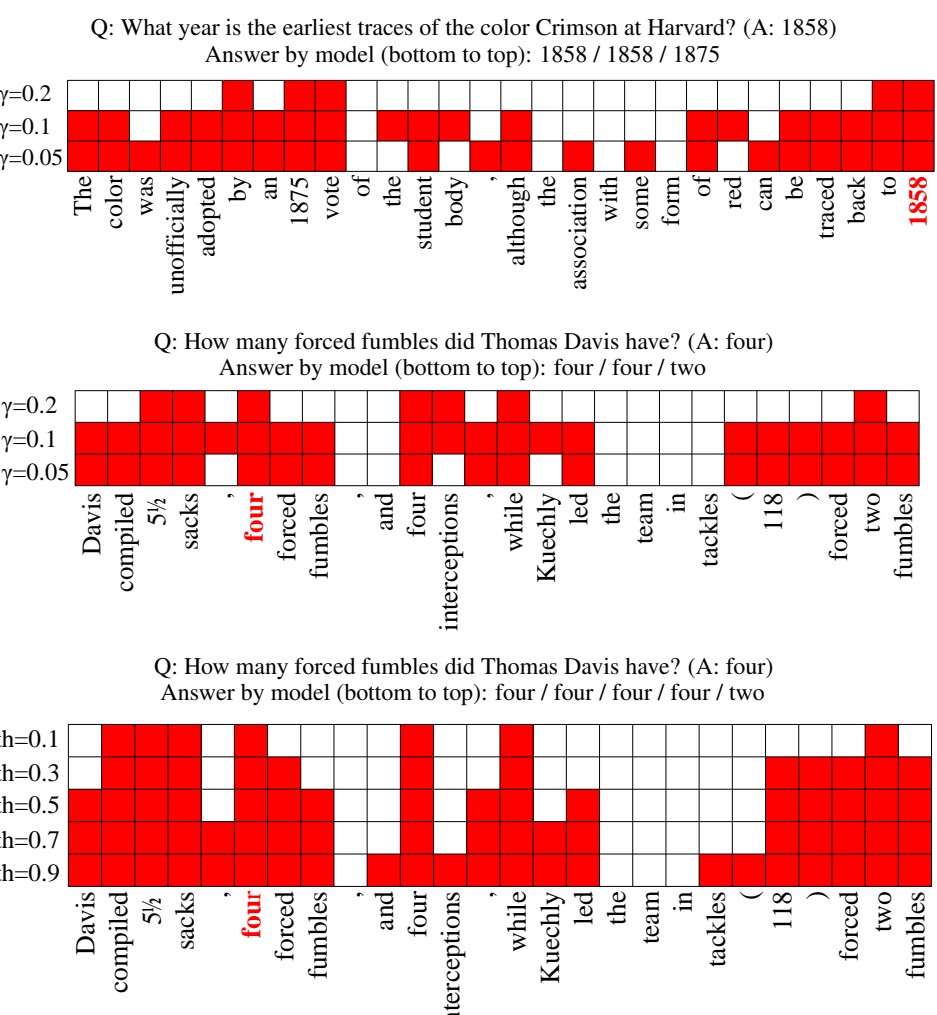

Figure 10: Reading (red) and skimming (white) on SQuAD, with LSTM+Attention model. The top two are skimming models with different values of $\gamma$. The bottom one is skimming models with different values of skim decision threshold (whose default is 0.5). An increase in $\gamma$ and a decrease in threshold lead to more skimming.

