# OpenReview forum: "Neural Speed Reading via Skim-RNN"
_ICLR.cc/2018/Conference — Accept (Poster)_

### Official Review · AnonReviewer1 · 2017-11-27
**Neural Speed Reading via Skim-RNN**

**Rating:** 7
**Confidence:** 3

**Review:**

The paper proposes a way to speed up the inference time of RNN via Skim mechanism where only a small part of hidden variable is updated once the model has decided a corresponding word token seems irrelevant w.r.t. a given task. While the proposed idea might be too simple, the authors show the importance of it via thorough experiments. It also seems to be easily integrated into existing RNN systems without heavy tuning as shown in the experiments.

* One advantage of proposed idea claimed against the skip-RNN is that the Skim-RNN can generate the same length of output sequence given input sequence. It is not clear to me whether the output prediction on those skimmed tokens is made of the full hidden state (updated + copied) or a first few dimensions of the hidden state. I assume that the full hidden states are used for prediction. It is somehow interesting because it may mean the prediction heavily depends on small (d') part of the hidden state. In the second and third figures of Figure 10, the model made wrong decisions when the adjacent tokens were both skimmed although the target token was not skimmed, and it might be related to the above assumption. In this sense, it would be more beneficial if the skimming happens over consecutive tokens (focus on a region, not on an individual token).

* This paper would gain more attention from practitioners because of its practical purpose. In a similar vein, it would be also good to have some comments on training time as well. In a general situation where there is no need of re-training, training time would be meaningless, however, if one requires updating the model on the fly, it would be also meaningful to have some intuition on training time.

* One obvious way to reduce the computational complexity of RNN is to reduce the size of the hidden state. In this sense, it makes this manuscript more comprehensive if there are some comparisons with RNNs with limited-sized hidden dimensions (say 10 or 20). So that readers can check benefits of the skim RNN against skip-RNN and small-sized RNN.

---

> ### Author Response · Authors · 2018-01-02
> **Our response**
>
> Thank you for your insightful and supportive comments; we make  a few clarifications and discuss additional experiments inspired by your suggestions.
>
> Clarification:
> - Output of skimmed tokens: The output of skimmed tokens is the full hidden state (concatenating updated and copied parts).
>
> Suggestions:
> - Focusing on region: Thank you for your suggestion, and we will consider this approach in future work.
>
> - Training time: Since Skim-RNN needs to compute outputs for both RNNs (big and small) during training, it requires more time for the same number of training steps. For instance, in SQuAD, Skim-LSTM takes 8 hours of training whereas LSTM takes 5 hours until convergence. However, in terms of number of training steps, they both require approximately 18k steps. We will include this in any final version of the paper.
>
> - Comparison with small hidden size: When the hidden size becomes 10 (from 100) for vanilla RNN, there is 3.4% accuracy drop in SST (7.1x less FLOP) and 6.1% accuracy drop in Rotten Tomatoes (7.1x less FLOP). There is a clear trade-off between accuracy and FLOP when smaller hidden size is used. We are currently experimenting with other datasets and will include them in the next revision.

---

### Official Review · AnonReviewer2 · 2017-11-27
**A model that makes intuitive sense, with solid experimentation**

**Rating:** 7
**Confidence:** 3

**Review:**

Summary: The paper proposes a learnable skimming mechanism for RNN. The model decides whether to send the word to a larger heavy-weight RNN or a light-weight RNN. The heavy-weight and the light-weight RNN each controls a portion of the hidden state. The paper finds that with the proposed skimming method, they achieve a significant reduction in terms of FLOPS. Although it doesn’t contribute to much speedup on modern GPU hardware, there is a good speedup on CPU, and it is more power efficient.

Contribution:
- The paper proposes to use a small RNN to read unimportant text. Unlike (Yu et al., 2017), which skips the text, here the model decides between small and large RNN.

Pros:
- Models that dynamically decide the amount of computation make intuitive sense and are of general interests.
- The paper presents solid experimentation on various text classification and question answering datasets.
- The proposed method has shown reasonable reduction in FLOPS and CPU speedup with no significant accuracy degradation (increase in accuracy in some tasks).
- The paper is well written, and the presentation is good.

Cons:
- Each model component is not novel. The authors propose to use Gumbel softmax, but does compare other gradient estimators. It would be good to use REINFORCE to do a fair comparison with (Yu et al., 2017 ) to see the benefit of using small RNN.
- The authors report that training from scratch results in unstable skim rate, while Half pretrain seems to always work better than fully pretrained ones. This makes the success of training a bit adhoc, as one need to actively tune the number of pretraining steps.
- Although there is difference from (Yu et al., 2017), the contribution of this paper is still incremental.

Questions:
- Although it is out of the scope for this paper to achieve GPU level speedup, I am curious to know some numbers on GPU speedup.
- One recommended task would probably be text summarization, in which the attended text can contribute to the output of the summary.

Conclusion:
- Based on the comments above, I recommend Accept

---

> ### Author Response · Authors · 2018-01-02
> **Our response**
>
> Thank you for your insightful and supportive comments; we discuss additional experiments following your suggestions and make a few clarifications.
>
> Suggestions:
> - Other gradient methods: Thank you for your suggestion, and we found that REINFORCE substantially underperforms (less than 20% accuracy on SST) Gumbel-Softmax within 50k steps of training. We suspect that this is due to the high variance of REINFORCE, which becomes even worse in our case where the sample space exponentially increases with the sequence length. We found that temperature annealing is not as bad as REINFORCE, but the accuracy is still ~0.5% lower than Gumbel-Softmax and the convergence is slower. We will include this in any final version of the paper.
>
>
> - Text summarization: We agree that it is an appropriate application for Skim-RNN. We will consider it for potential future work.
>
>
> Clarifications:
> - Adhoc training due to required pretraining: while pretraining definitely helps in QA, we would like to emphasize that no-pretraining still performs well (classification results are without pretraining), and there is no added cost of pretraining; that is, pretraining + finetuning has a similar training time to training from scratch.
>
> - GPU speed up: Theoretically Skim-RNN could have speed up on GPU. However,  because parallelization has log-time cost, this would be negligible compared to other costs.

---

### Official Review · AnonReviewer3 · 2017-11-27
**A review of "Neural Speed Reading via Skim-RNN"**

**Rating:** 8
**Confidence:** 3

**Review:**

This paper proposes a skim-RNN, which skims unimportant inputs with a small RNN while normally processes important inputs with a standard RNN for fast inference.

Pros.
-	The idea of switching small and standard RNNs for skimming and full reading respectively is quite simple and intuitive.
-	The paper is clearly written with enough explanations about the proposal method and the novelty.
-	One of the most difficult problems of this approach (non-differentiable) is elegantly solved by employing gumbel-softmax
-	The effectiveness (mainly inference speed improvement with CPU) is validated by various experiments. The examples (Table 3 and Figure 6) show that the skimming process is appropriately performed (skimmed unimportant words while fully read relevant words etc.)
Cons.
-	The idea is quite simple and the novelty is incremental by considering the difference from skip-RNN.
-	No comments about computational costs during training with GPU (it would not increase the computational cost so much, but gumbel-softmax may require more iterations).

Comments:
-	Section 1, Introduction, 2nd paragraph: ‘peed’ -> ‘speed’(?)
-	Equation (5): It would be better to explain why it uses the Gumbel distribution. To make (5) behave like argmax, only temperature parameter seems to be enough.
-	Section 4.1: What is “global training step”?
-	Section 4.2, “We also observe that the F1 score of Skim-LSTM is more stable across different configurations and computational cost.”: This seems to be very interesting phenomena. Is there some discussion of why skim-LSTM is more stable?
-	Section 4.2, the last paragraph: “Table 6 shows” -> “Figure 6 shows”

---

> ### Author Response · Authors · 2018-01-02
> **Our response**
>
> Thank you for your insightful and supportive comments; we discuss additional experiments inspired  your suggestions and make a few clarifications.
>
>
> Suggestions:
> - Training cost with GPU: Thank you for the suggestion, and we report training cost in two dimensions: memory and time. Assuming d/d’=100/20 on SQuAD, memory consumption is only ~5% more than vanilla RNN. Since Skim-RNN needs to compute outputs for both RNNs (big and small) during training, it requires more time for the same number of training steps. For instance, on SQuAD, Skim-LSTM takes 8 hours of training whereas LSTM takes 5 hours until convergence. However, in terms of number of training steps, they both require approximately 18k steps.
>
> - Why Gumbel-softmax and not just temperature: we used Gumbel-Softmax mainly due to its theoretical guarantee shown in Jang et al (2017). We experimented with temperature annealing only, and found that the accuracy is ~0.5% lower on SQuAD and convergence is a little slower. While there is some advantage of Gumbel-softmax, It seems temperature annealing is also an effective technique.
>
> - Typos: thank you for correcting them and we have fixed them in the current revision.
>
>
> Clarifications:
> - Sec 4.1 global training step: We meant just “training step”, and we fixed it in the most recent revision.
>
> - Sec 4.2 stableness: We actually meant that LSTM accuracy dips (is unstable) when we use smaller hidden state size to reduce FLOP, while Skim-LSTM accuracy does not dip even with high reduction in FLOP.

---

### Public Comment · (anonymous) · 2017-11-10
**Skimming part of the proposed model**

It is a very interesting paper. I really enjoyed reading it.  Actually, I have a naive question about skimming part. According to Figure 1, the hidden state (d’ + 1 to d) of the word “and” is copied from the previous hidden state directly, while the first part (1 to d’) is updated by a smaller RNN. I am curious about this setting. Why did you update this model in this way? Can I copy the (1 to d - d’) part and update the remaining part of this model?

---

> ### Author Response · Authors · 2017-11-10
> **Response**
>
> Hi,
> Thank you for your interest in our paper and your suggestion!
> I think copying 1 to d-d' all the time is equivalent to copying d'+1 to d (without loss of generality).
> So I believe what you are suggesting is something like having two small RNNs that operate on different parts of the hidden state.
> In fact, we also think it is an interesting direction to explore (we also mention this as potential future work in the conclusion), though we did not report it mainly because we have not observed clear advantage by doing so in our experiments yet.

---

> > ### Public Comment · (anonymous) · 2017-11-11
> > **Response**
> >
> > Hi,
> > Thanks for your response.

---

### Comment · Area_Chair · 2018-01-09
**extremely similar idea from ICLR'17**

Note: this is not an official meta-review

the idea in this paper looks very similar to the idea from <VARIABLE COMPUTATION IN RECURRENT NEURAL NETWORKS> which was presented at ICLR'17: https://arxiv.org/abs/1611.06188. Especially, looking at Fig. 1's of both papers clearly indicate the similarities between these two approaches.

i'd like the authors to clarify how they differ, and would like to ask the reviewers to read https://arxiv.org/abs/1611.06188 and see how this affects your judgement of the submission.

---

> ### Author Response · Authors · 2018-01-10
> **Difference between VCRNN and Skim-RNN**
>
> Thank you for mentioning a relevant paper that we missed!
>
> We agree that both of Skim-RNN and VCRNN (as well as LSTM-Jump) are concerned with dynamically controlling the computational cost of RNN, and we will make sure to discuss VCRNN in our next revision. However, we would like to emphasize that there is a fundamental difference between them: VCRNN partially updates the hidden state (controlling the number of units to update at each time step), while Skim-RNN contains multiple RNNs that “share” a common hidden state with different regions on which they operate (choosing which RNN to use at each time step). This has two important implications.
>
> First, the nested RNNs in Skim-RNN have their own weights and thus can be considered as independent agents that interact with each other through the shared state. That is, Skim-RNN updates the shared portion of the hidden state differently (by using different RNNs) depending on importance of the token, whereas the affected (first few) dimensions in VCRNN are identically updated regardless of the importance of the input. We argue that this capability of Skim-RNN could be a crucial advantage, based on the following initial observation. Instead of having two independent nested RNNs, we experiment with a single RNN and a binary decision function whether to update the hidden state fully (100 dimensions) or partially (first 5 dimensions). On SST, “partial update” model (similar to VCRNN but binary decision instead) underperformed Skim-RNN by 2.0% with similar skimming rate.
>
> Second, at each time step, VCRNN needs to make a d-way decision (where d is the hidden state size, usually hundreds), whereas Skim-RNN only requires binary decision. This means that computing exact gradient of VCRNN is even more intractable (d^L vs 2^L) than that of Skim-RNN, and subsequently the gradient estimation would be harder as well.
>
> Experimentally, the two papers focus on different domains. VCRNN experimented on one music modeling task and and two (bit and char level) language modeling tasks, while we experimented on four language classification tasks and two question answering tasks. We would like to also appeal that the reviewers have acknowledged our diverse experiments, analyses, and visualizations that are useful to understand, verify and interpret our model, which we believe is a meaningful contribution towards the community’s effort on reducing the computational cost of RNNs.
>
> As it is past the rebuttal period, we would like to know if we can make a revision to the submission, and if so when the deadline is. In the meanwhile, we will do our best to update the paper and/or provide additional results as soon as possible with VCRNN considered.

---

> > ### Comment · Area_Chair · 2018-01-10
> > **experiments requested**
> >
> > thanks for the detailed description, but they still do look quite similar. the "partial update" model is also not exactly what VCRNN does, in the sense that it's a very much crippled version of VCRNN without, e.g., saving any computation. it'll be important to carefully compare the full implementation of VCRNN against the skim-RNN on at least one task. after all, the VCRNN was proposed in the *same* venue just one year ago.
> >
> > please feel free to make another revision however as early as you could.

---

> > > ### Author Response · Authors · 2018-01-15
> > > **Experiments on SST**
> > >
> > > Thank you for the comment, and as you suggested, we report a comparison between Skim-RNN and VCRNN on Stanford Sentiment Treebank (SST). Since VCRNN’s accuracy had a high variance, we ran the experiment 5 times for both models with different random initialization of the weights. Skim-RNN (d=100, d’=5) obtained an average accuracy of 85.6% (std=0.47%, max=86.4%) with average FLOP reduction of 2.4x, while VCRNN obtained an average accuracy of 81.9% (std=4.91%, max=85.7%) with average FLOP reduction of 2.6x. So there is a clear advantage of Skim-RNN over VCRNN on average accuracy (3.7% diff), max accuracy (0.7% diff), and stability (std) with similar FLOP reduction. We will be working on the rest of the tasks and make sure to report the results in a future revision.

---

### Author Response · Authors · 2018-01-24
**Revised on Jan 24**

In addition to the SST experiment that we reported previously, we also just finished experimenting VCRNN on SQuAD.
Using RNN+Att explained in the paper as a base model (where the RNN is replaced with either Skim-LSTM or VCRNN), VCRNN obtained F1=74.9% and EM=65.4% with very little FLOP reduction (less than 1.01x), which is worse than F1=75.7% and EM=66.7% with the FLOP reduction of 1.4x by Skim-LSTM.

We made a revision to the paper that includes these discussions and the experimental results of VCRNN.

EDIT on Jan 25: We reported a wrong number for VCRNN's FLOP reduction; it should be <1.01x, not 1.4x.

---

> ### Author Response · Authors · 2018-01-25
> **Note on VCRNN's FLOP reduction on SQuAD**
>
> We note that we could not increase FLOP reduction of VCRNN by controlling the hyperparameters on SQuAD. Also, VCRNN performs worse  than vanilla RNN (LSTM) without any gain in FLOP reduction, which we believe is due to the difficulty in training (biased gradient, etc.).
>
> We believe that this supports our claim that Skim-RNN has some crucial advantages over VCRNN that we discussed in our previous comment (and in the related work of the current revision).

---

> > ### Comment · Area_Chair · 2018-01-25
> > **thank you for the update**
> >
> > NT

---

### Decision · Program_Chairs · 2018-01-29
**ICLR 2018 Conference Acceptance Decision**

**Decision:**

Accept (Poster)

**Comment:**

this submission proposes an efficient parametrization of a recurrent neural net by using two transition functions (one large and one small) to reduce the amount of computation (though, without actual improvement on GPU.) the reviewers found the submission very positive.

please, do not forget to include all the result and discussion on the proposed approach's relationship to VCRNN which was presented at the same conference just a year ago.